# Full-Length Genome and Partial Viral Genes Phylogenetic and Geographical Analysis of Dengue Serotype 3 Isolates

**DOI:** 10.3390/microorganisms9020323

**Published:** 2021-02-04

**Authors:** Muhammad Amir, Abrar Hussain, Muhammad Asif, Sagheer Ahmed, Hina Alam, Marius Alexandru Moga, Maria Elena Cocuz, Luigi Marceanu, Alexandru Blidaru

**Affiliations:** 1Department of Biotechnology, BUITEMS, Baluchistan University of Information & Technology, Engineering and Management Sciences (BUITEMS), Quetta 87300, Pakistan; sky_amir786@yahoo.com (M.A.); abrarbangash176@hotmail.com (A.H.); asifjallali@hotmail.com (M.A.); 2Office of Research Innovation and Commercialization, Baluchistan University of Information & Technology, Engineering and Management Sciences (BUITEMS), Quetta 87300, Pakistan; 3Shifa College of Pharmaceutical Sciences, Shifa Tameer-e-Millat University, Islamabad 44000, Pakistan; sagheer.scps@stmu.edu.pk; 4Pakistan Institute of Medical Sciences, Islamabad 44000, Pakistan; hina_alam@icloud.com; 5Faculty of Medicine, Transilvania University of Brasov, 500019 Brasov, Romania; moga.og@gmail.com; 6Faculty of Medicine, University of Medicine and Pharmacy Carol Davila, 020021 Bucharest, Romania; alexandrublidaru@yahoo.com

**Keywords:** dengue virus, dengue hemorrhagic fever, phylogenetic analysis, maximum likelihood

## Abstract

Dengue fever is among the most common vector-borne diseases. Dengue virus (DENV), responsible for dengue fever as well as dengue hemorrhagic fever, belongs to the genus flavivirus and family Flaviviridae. Flaviviruses infect various vertebrate species and arthropods and are also responsible for diseases in birds, wild animals, and primates. DENV consists of a single-stranded, positive-sense RNA genome ~11 kb in size. Complete genome and partial gene sequences of geographically distinct DENV-3 strains were retrieved from the GenBank database. The evolutionary divergence of the 33 whole-genome and individual gene sequences of the nucleotides and amino acids of DENV-3 strains were generated with the maximum likelihood (ML) and Bayesian phylogenetic study (BEAST) methods using the MEGA 7 software. The genome size varied from 10,484 to 10,724 nucleotides among the strains with distinct geographical backgrounds belonging to Central America, South-Central Asia, and Eastern Asia. A phylogenetic analysis of the nucleotide and amino acid sequences of these DENV-3 isolates revealed extensive differences in the topologies due to PrM/M, NS1, NS2B, and NS3 genes. These results suggest substantial variation in the evolutionary pathways of the studied genes and genomes.

## 1. Introduction

Dengue fever (DF) is one of the most common vector-borne diseases in the world, mostly found in tropical as well as in subtropical regions. Dengue virus (DENV), the causative microorganism of DF and dengue hemorrhagic fever (DHF), is a member of the Flavivirus genus and belongs to the Flaviviridae family. All flaviviruses contain single-stranded, positive-sense RNA and mostly infect hematophagous arthropods (ticks or mosquitoes), which complements its natural horizontal transmission cycle [1]. Infection with flaviviruses varies from asymptomatic to lethal, and more than 50% of viruses may cause human diseases. Influenza-like illness with sudden onset of fever, arthralgia, myalgia, retro-orbital headaches, maculopapular rash, leukopenia, vascular leakage, and encephalitis are mostly associated with flaviviruses infections [2]. The genus flavivirus also contains several important disease-causing viruses such as yellow fever virus (YFV), West Nile virus (WNV), and Japanese encephalitis virus (JEV), as well as tick-borne encephalitis virus (TBEV), associated with tick-borne diseases [3].

DENV serotypes have been classified into DENV 1–4, each possessing distinct antibody epitopes specific to the serotype [4]. Previous studies show that these four DENV serotypes evolved from their common ancestor in separate ecological niches. Each ecological niche was different, with a specific geographic location, distinct primate host populations, and different vector species [5]. The dengue virus genome is ~11 kb. It comprises an open reading frame (ORF), which codes for ten mature proteins, of which three are structural proteins (capsid (C), premembrane or membrane (prM/M), envelope (E)) and seven are nonstructural (NS) proteins (NS1, NS2A, NS2B, NS3, NS4A, NS4B, and NS5), many of which play important roles in replication of the virus. The most immunologically significant protein is envelope (E) glycoprotein [6].

In around 1780, the first DHF epidemics were reported in Asian, African, and North American regions. For approximately the next 200 years, DHF epidemics remained localized in these three continents, albeit the epidemics seemed to be concentrated and more frequent in the tropics [7]. DENV-1 and DENV-2 were isolated during the Second World War in Japanese and American patients. DENV-3 and DENV-4 were isolated in the 1950s during the outbreaks in the Philippines and Thailand, respectively [8]. Infection caused by DENV may be clinical or subclinical, mild or acute, or afebrile or febrile, which may become severe and result in hemorrhagic disease [9]. In most cases, it causes DF, with symptoms including frontal headache, fever, myalgias, nausea, vomiting, rash, and often arthralgias.

DHF and dengue shock syndrome (DSS), a vascular leak disease initiated by the overwhelming of the immune system by the cells of the monocytic descent, are two of the most severe forms of DENV infection. In the most severe cases, infection with DENV will cause substantial hemorrhages, organ failure, and neural disorder that impersonate viral encephalitis [10]. The treatment of dengue fever is symptomatic, while DSS and DHF need fluid resuscitation treatment [6]. Based on genomic diversity, DENV serotypes are further classified into various subtypes/genotypes [11]. DENV-3 is further divided into genotypes 1 to 5 [12]. Most DENV-3 infections are caused by genotypes 1 and 3, with most of the DF and DHF outbreaks in the Indian subcontinent, the Americas, Southeast Asia, East Africa, and the South Pacific being due to this genotype, whereas genotypes 4 and 5 were not implicated with DHF epidemics [13].

The most commonly used technique for DENV genotyping employs the phylogenetic study of gene structures, specifically the E gene [14]. Previous phylogenetic studies in each of the four different DENV serotypes revealed genetic subtypes that differ up to 12% in the nucleotide sequence in the E gene, which is known to be the most antigenic part of the virus [15]. In the current study, we performed a comparative phylogeographic and phylogenetic analysis of 33 full genomes as well as individual gene sequences of DENV-3 strains to identify genetic variations and their evolutionary patterns over time.

## 2. Materials and Methods

### 2.1. Sequence Analysis and Phylogenetic Hierarchy

The complete genomes of 33 DENV-3 strains were downloaded from GenBank (table). These genomes were selected from a distinct topographical background surrounding the Central American, South-Central Asian, and Eastern Asian regions, where dengue outbreaks have previously been reported. Complete genome sequences from each strain were manually separated into 12 fragments. These fragments include 5′ and 3′ untranslated regions, three structural genes (capsid (C), premembrane or membrane (prM/M), envelope (E)), and seven nonstructural genes (NS1, NS2A, NS2B, NS3, NS4A, NS4B, and NS5). Nucleotide sequences of all partial genes, 5′ and 3′ untranslated regions (UTRs), and complete genome nucleotide as well as amino acid sequences were aligned using the Clustal W program [16]. Phylogenies were generated using the MEGA7 program based on general time-reversible (GTR)/GTR + I + G nucleotide substitution models of maximum likelihood (ML). The strengths of phylogenies were evaluated by resampling with 1000 bootstrap replications.

### 2.2. Evolutionary Length among DENV-3 Strains

After the sequences of complete genomes or partial genes were aligned and analyzed, the same was further analyzed to determine the comprehensive evolutionary length between the DENV-3 strains. This was also carried out using the MEGA7 program [17]. The bootstrap resampling review was achieved via 1000 repetitions (Table 1).

## 3. Results

The 33 complete genome DENV-3 strains ranged in size from 10,484 to 10,724 nucleotides with various topographical backgrounds, including Central America, South Central Asia, and Eastern Asia. However, the length of the open reading frame (ORF) at ~10,240 was similar across the DENV-3 strains (Table 1). The key variance in the genome lengths was found in the 5′ or 3′ UTRs rather than in the ORFs.

### 3.1. Phylogeneses and Phylogeographic Presentation

The evolutionary study of DENV viral and topographical genes showed the processes by which the four serotypes (DENV1-4) differed and independently entered the human population, along with the locations in which the viral populations persisted for the four serotypes. This model, which depicted DENV phylogenetic antiquities, was built from a set of observed viral structures by employing maximum likelihood and Bayesian evolutionary analysis.

Our study model, which was used to reconstruct the evolutionary history of gene sequences, relies on sequence similarity in the form of a phylogenetic tree with individual nodes. Each of these individual nodes represents the most conjectural, contemporary common ancestor of the observed gene structures located at the tips of the trees. The maximum likelihood method chooses the tree that best explains the evolutionary pathway of the provided sequence data.

Our analysis, performed in MEGA7, revealed the presence of five clusters, namely Oceania, Southeastern Asia, Central America, South-Central Asia, and Eastern Asia, which are part of two main clusters: Oceania and the rest of the world (Figure 1). The Oceania cluster consists of Southeastern Asia, whereas the Philippines (Accession No.: KM190937) was found away from both of these clusters.

### 3.2. Bayesian Evolutionary Analysis (BEAST) of Whole-Genome DENV-3

Figure 2 describes the same main clusters as mentioned in Figure 1. The first subcluster consists of Australia (Accession No.: JN406515), Singapore (Accession No.: GU370052), Indonesia (Accession No.: KC762693), and East Timor (Accession No.: AB214879). The second subcluster consists of Samoa (Accession No.: FJ898456), Cook Island (Accession No.: FJ898455), French Polynesia (Accession No.: JQ920479), Wallis and Futuna (Accession No.: JQ920489), and the Philippines (Accession No.: KM190937).

The second main cluster comprises China (Accession No.: KF824902), Mexico (Accession No.: FJ898442), and the USA (Accession No.: FJ182010), up to Grenada (Accession No.: KF955505). The origin of Pakistan (Accession No.: KF041254) and India (Accession No.: GQ466079) is rooted at the same node in this cluster.

### 3.3. Phylogenetic Analysis Based on Amino Acids

Phylogenetic analysis based on amino acids was applied using the ML approach with reference to the JTT matrix-based pattern with the highest log likelihood (‒13480.8871) (Figure 3). In this analysis, Eastern Asia (China (Accession No.: KF824902), Vietnam (Accession No.: FJ547066), Thailand (Accession No.: FJ687448), and Cambodia (Accession No.: FJ639715)) moved towards the Oceania cluster. This was a key difference between the whole-genome nucleotide analysis (Figure 1) and amino acid phylogenetic presentation (Figure 3).

### 3.4. Phylogenetic Analysis of Capsid (C) and Envelope (E) protein

Changes in the amino acid sequences of the C protein moved from the Philippines strain (Accession No.: AIG60036) into the Eastern Asian cluster, whereas E protein changes moved from the China strain (Accession No.: AHI17474) near the Philippines strain (Accession No.: AIG60036) and away from the East Asian cluster (Figure 2, Figure 3, Figure 4 and Figure 5A).

### 3.5. Phylogenetic Analysis Based on Individual Proteins

Phylogenetic examination of the PrM/M gene shows a large deviation in the topologies of the first and second main clusters based on the analysis of individual genes (nucleotide) and their respective amino acid analysis (Figure 4B and Figure 5B). The disagreement between the nucleotide and amino acid analysis of the NS1 gene resulted in the shifting of some strains in the subgroups of the first main cluster to the second cluster (Figure 4D and Figure 5D). The NS2B analysis illustrates the significant deviation of the subgroups of the first and second main clusters (Figure 4F and Figure 5F). Similarly, a comparison of the NS3 nucleotide and amino acid analysis shows topological deviations between the subgroups of the main clusters, resulting in the Eastern Asian falling into the second main cluster (Figure 4G and Figure 5G). Unlike C, E, PrM/M, NS1, NS2B, and NS3, the nucleotide and amino acid analysis of other genes such as NS4A and NS4B, as well as NS5, did not show significant deviations and movements between the two clusters (Figure 4H–J and Figure 5H–J).

## 4. Discussion

Previous studies have employed individual gene sequences of DENV-3 for phylogenetic investigations and classified them into various genotypes [18]. Most of such studies use E gene sequences for the genotyping of DENV-3 [19], although other genes have been used by some investigators [20]. For example, based on the phylogenetic analysis of E gene sequences, DENV-3 was split into four genotypes [21]: strains from South Pacific islands and Southeast Asia fall into genotype 1; genotype 2 comprises of strains from Thailand; genotype 3 consists of strains from East Africa, Samoa, and the Indian subcontinent; and strains from Puerto Rico Tahiti fell into genotype 4 [12]. Therefore, earlier studies were more focused on the sequence analysis of individual genes or a subset of those genes for genotyping of DENV-3 [22], especially the PrM/M, E, NS1, NS3, NS4A, and NS5 genes [23]. For example, during the 2015 epidemic of DENV in the province of Khyber Pakhtunkhwa, Pakistan, a phylogenetic analysis, made on the basis of the E and S1 genes, revealed that it belongs to DENV-3, showing maximum homology with strains previously reported from Pakistan and India [23]. This practice, although useful, is based on limited information, in most cases merely restricted to domestic sequences in one territory. However, more recently, investigators are using entire genome nucleotide or amino acid sequences for identifying genetic variation, constructing phylogenetic trees, and the classification of DENV genotypes. A recent study employing the entire genome of DENV-3 (10,672 bp) identified 388 mutations in the nucleotide sequences and 34 mutations in the amino acids [24].

Our investigation, by analyzing entire genomes of 33 DENV strains from around the world, confirmed the previous findings of Pakistani strains of DENV being placed in the DENV-3 genotype. Previous studies utilizing the complete genome sequences of DENV-3 have yielded similar results [25]. However, our investigation utilizing the entire genome yields additional important information. In particular, our analysis of amino acid sequences of the entire ORF produced some unexpected results, which is why we see several DENV strains swapping clusters after amino acid sequence analysis.

Previously, phylogenetic correlations using whole-genome sequences of Indian DENV-3 isolates described the presence of a discrete DENV-3 clade in India [26]. In the present study, our results confirmed the distinctiveness of the Indian DENV-3 isolates in the South-Central Asia subcluster through the analysis of complete genome nucleotide and amino acid sequences.

Phylogenetic trees generated from complete genomes using the ML technique presented two main clusters Oceania and the rest of the world (Figure 1). When the nucleotide sequences of the individual genes and their respective protein sequences were analyzed via p-blast, the topology network of the genomes changed considerably (Figure 4 and Figure 5). When the analysis was carried out based on the C protein, DENV strains from countries such as Taiwan, the USA, Peru, Sri Lanka, Pakistan, Brazil, and India concentrated on a single node but were previously located on multiple nodes when the analysis was performed using nucleotide sequences of the C gene (Figure 4A and Figure 5A). In the Oceania cluster, constructed based on the PrM/M gene, there are around nine countries, but due to changes in the PrM/M, protein only Singapore, Australia, and Indonesia are left in the cluster (Figure 4B and Figure 5B). Similarly, for many other genes, strains from several countries changed clusters when phylogenetic trees were constructed based on amino acid sequences (Figure 4 and Figure 5).

Before the start of this investigation, one of our assumptions was that phylogenetic analysis based on nucleotide and amino acid analysis yields the same results. However, our analysis with phylogenetic trees based on both nucleotide and amino acid sequences did not produce the same results. This might be due to the redundancy of the codons. Changes at the nucleotide level are sometimes not reflected at the amino acid level. Our analysis based on amino acid sequences shows a different phylogenetic tree from the one which we constructed with nucleotide sequences (Figure 1 and Figure 2). Some of the strains swapped clusters when the phylogenetic tree was constructed based on amino acid sequences.

Genomic regions, which displayed significant variability in our investigation and showed interesting, results include the PrM/M (Figure 4B and Figure 5B), NS1 (Figure 4D and Figure 5D), NS2B (Figure 4F and Figure 5F), and NS3 genes (Figure 4G and Figure 5G). These genes exhibited substantial variability both at the nucleotide and amino acid levels. In particular, phylogenetic analysis of the PrM/M gene showed large variability and, as a result, changed the topologies of certain strains between the first and second main clusters (Figure 4B and Figure 5B). Some of the genes such as NS4A, NS4B, and NS5 did not show significant variation (Figure 4H–J and Figure 5H–J).

All the genes that improve viral fitness to survive and reproduce will keep changing to achieve even better functionality. Viral fitness depends more on certain genes than others. It has been reported previously that amino acid differences at position 390 of the E protein and in the 5′ and 3′ UTRs confer higher efficiency for viral replication in monocyte-derived dendritic cells and macrophages and thus improve the fitness of the virus [27]. On the other hand, the genes that do not confer survival advantage or are functionally constrained are less likely to change. Previous studies show that some of the nonstructural proteins such as NS3 and NS5 have limitations on how much they can change without compromising their function [28].

NS5 gene coding for an RNA-dependent RNA polymerase did not display significant variability in our analysis. Viral polymerases, like other important structural and nonstructural proteins, are under enormous selection pressure. However, unlike many structural proteins, enzymes cannot change too much so as not to compromise their functions. Viral polymerases are also constrained by the fact that functional redundancy for polymerases is much less than for other enzymes, further putting them under pressure. Therefore, due to functional limitations, polymerases are usually highly conserved with occasional changes that would further increase their fitness for viral replication.

Our extensive analysis of viral mutations and evolution from 33 complete viral genomes (DENV-3 isolates) shows substantial differences from previous results, which were based on the nucleotide and amino acid sequences of individual genes (E, PrM/M, NS1, NS2B, and NS3) [29]. The topologies of various genes become significantly different when whole-genome nucleotide as well as amino acid sequences are analyzed instead of individual genes. These results suggest that the evolution of the DENV-3 genome is guided by both structural as well as in nonstructural parts of the genome.

## 5. Conclusions

Our study shows that a phylogenetic analysis based on complete genome nucleotide sequences of the distinct DENV-3 genotypes from around the world produces phylogenetic trees, which are different from when the analysis is performed based on amino acid sequences. Our investigation also shows that both structural and nonstructural parts of the genome are likely to shape the evolution of the DENV-3 genome.

## Figures and Tables

**Figure 1 microorganisms-09-00323-f001:**
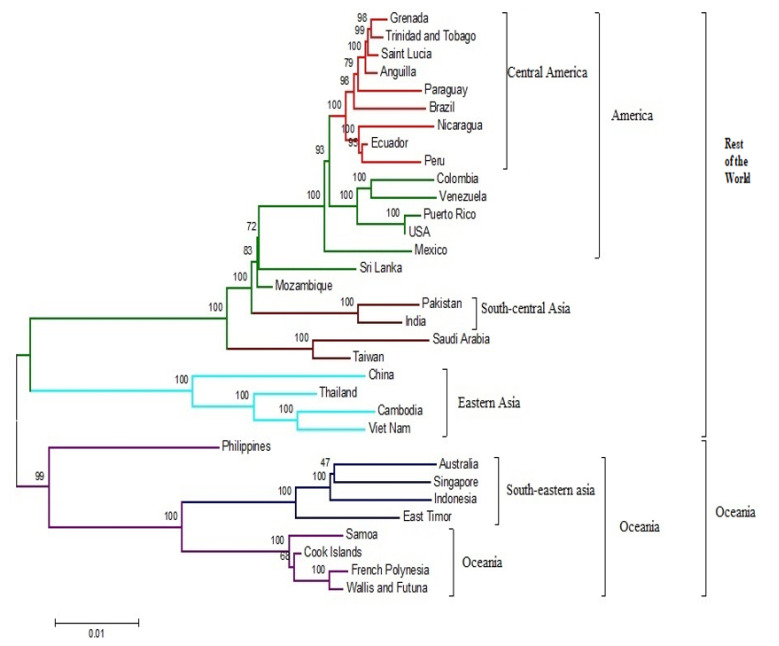
Phylogenetic maximum-likelihood tree of the DENV-3 full-genome nucleotide sequences. Trees were constructed using the MEGA v7 software with the bootstrap support of 1000 replicates. All nucleotide sequences were downloaded from the GenBank database for analysis as well as their respective DENV-3 isolates with their accession numbers listed.

**Figure 2 microorganisms-09-00323-f002:**
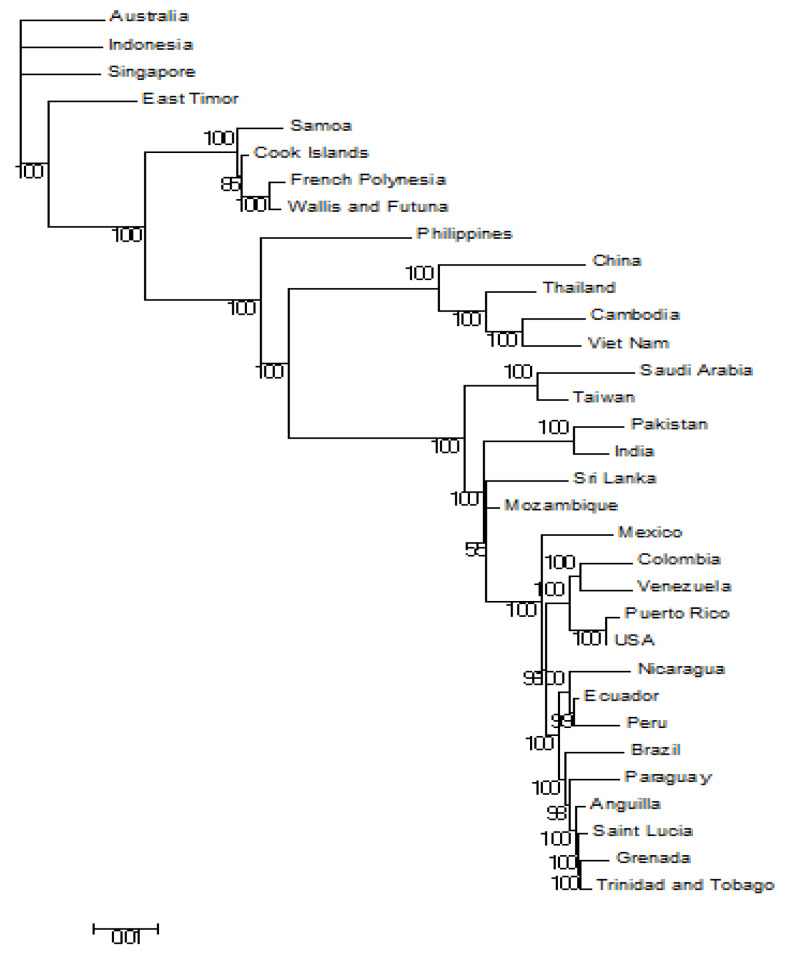
Bayesian evolutionary analysis (BEAST) of whole-genome DENV-3.

**Figure 3 microorganisms-09-00323-f003:**
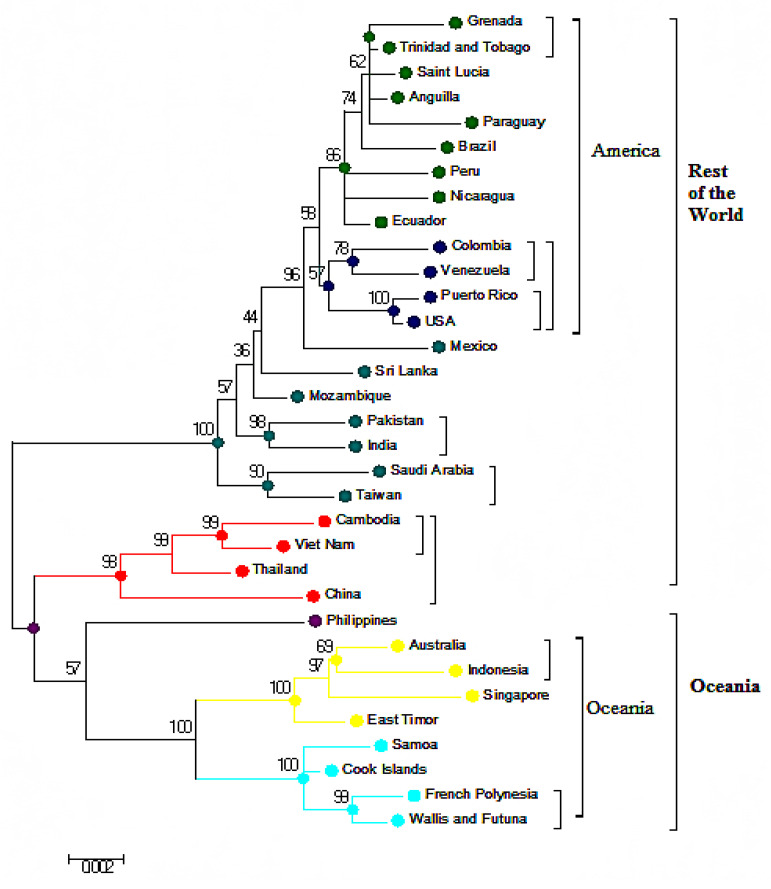
Phylogenetic maximum-likelihood tree of the DENV-3 full-genome amino acid sequences. Maximum-likelihood (ML) trees were constructed using the MEGA v7 software with the bootstrap support of 1000 replicates. All amino acid sequences were downloaded from the GenBank database for analysis with their accession numbers.

**Figure 4 microorganisms-09-00323-f004:**
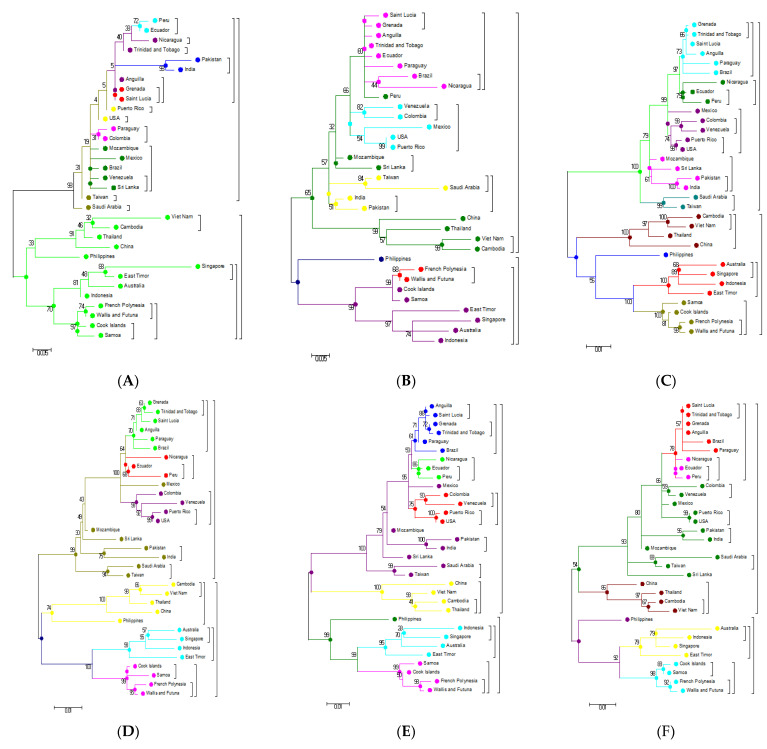
Molecular phylogenetic analysis using the maximum likelihood method based on the general time-reversible model of DENV-3 individual gene nucleotide sequences. All sequences of the individual genes were separated from the complete genome sequences that were downloaded from the GenBank database for analysis: (**A**) C gene; (**B**) PrM/M; (**C**) E gene; (**D**) NS1 gene; (**E**) NS2A; (**F**) NS2B; (**G**) NS3; (**H**) NS4A; (**I**) NS4B; (**J**) NS5; (**K**) 3′ UTR; (**L**) 5′ UTR.

**Figure 5 microorganisms-09-00323-f005:**
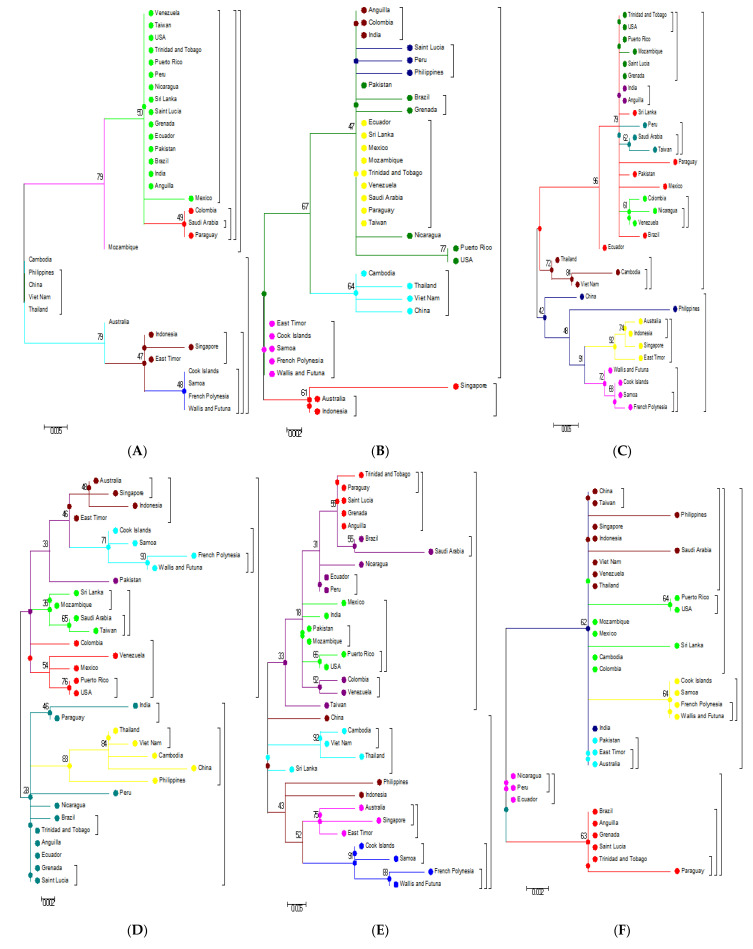
Molecular phylogenetic analysis using the maximum likelihood method based on the general time-reversible model of DENV-3 individual gene amino acid sequences. All sequences of the individual genes were separated from the complete genome sequences downloaded from the GenBank database for analysis. (**A**) C; (**B**) PrM/M; (**C**) E; (**D**) NS1; (**E**) NS2A; (**F**) NS2B; (**G**) NS3; (**H**) NS4A; (**I**) NS4B; (**J**) NS5.

**Table 1 microorganisms-09-00323-t001:** Locations and names of the DENV-3 isolates collected worldwide.

Country	Name of Isolate	Genome (bp)	Year	Nucleotide Accession	Protein Accession	Reference
Anguilla	AI/BID-V2976/2001	10,663	2001	FJ898462	ACQ44501	Direct Submission
Australia	Cairns 2008	10,707	2008	JN406515	AFN80339	Direct Submission
Cambodia	KH/BID-V2053/2008	10,648	2008	FJ639715	ACL99233	Direct Submission
Colombia	CO/BID-V3405/2007	10,659	2007	GQ868578	ACW83006	Direct Submission
East Timor	Hu/TL018NIID/2005	10,707	2005	AB214879	BAE48725	Direct Submission
Ecuador	EC/BID-V2975/2000	10,663	2000	FJ898457	ACQ44496	Direct Submission
French Polynesia	PF96/150296-46183	10,671	1994	JQ920479	AFY10043	Direct Submission
India	DEL-72	10,680	2008	GQ466079	ADM63678	Direct Submission
Mexico	MX/BID-V2989/2007	10,663	2007	FJ898442	ACQ44481	Direct Submission
Viet Nam	VN/BID-V1911/2008	10,637	2008	FJ547066	ACL98983	Direct Submission
Mozambique	MZ/BID-V2418/1985	10,663	1985	FJ882575	ACQ44384	Direct Submission
Paraguay	PAR 5532-07	10,707	2007	HQ235027	AEF65939	Direct Submission
Saint Lucia	LC/BID-V2979/2001	10,660	2001	FJ898463	ACQ44502	Direct Submission
Singapore	SGEHI(D3)0040Y09	10,250	2009/01	GU370052	ADC92353	Direct Submission
Sri Lanka	LK/BID-V2409/1997	10,682	1997	GQ252674	ACS32036	Direct Submission
Taiwan	99TW628	10,707	1999	DQ675533	ABG73599	Direct Submission
Thailand	TH/BID-V2318/2001	10,629	2001	FJ687448	ACN42695	Direct Submission
Trinidad and Tobago	TT/BID-V2982/2002	10,663	2002	FJ898459	ACQ44498	Direct Submission
USA	US/BID-V1620/2005	10,648	2005	FJ182010	ACH99657	Direct Submission
Venezuela	VE/BID-V2267/2008,	10,654	2008	FJ639826	ACL99113	Direct Submission
Samoa	WS/BID-V2973/1995	10,663	1995	FJ898456	ACQ44495	Direct Submission
Wallis and Futuna	WF95/090595-2448	10,671	9/5/1995	JQ920489	AFY10053	Direct Submission
Cook Islands	CK/BID-V2972/1991	10,663	1991	FJ898455	ACQ44494	Direct Submission
Brazil	BR/AL95/2009	10,707	2009	JF808120	AFK83755	Direct Submission
China	YN01	10,707	2013	KF824902	AHI17474	Direct Submission
Grenada	GD/BID-V3930/2002	10,653	2002	KF955505	AHG23270	Direct Submission
Indonesia	MKS-WS79b	10,707	29/03/2010	KC762693	AHG06377	Direct Submission
Nicaragua	NI/BID-V7658/2012	10,569	4/7/1905	KF973480	AHC98451	Direct Submission
Pakistan	Pakistan/56/2008	10,675	2008	KF041254	AHC72426	Direct Submission
Peru	PE/BID-V7289/2008	10,693	2008	KJ189301	AHI43684	Direct Submission
Philippines	VIROAF7	10,267	1964	KM190937	AIG60036	Direct Submission
Puerto Rico	PR/BID-V1728/2006	10,645	2006	KF955456	AHG23221	Direct Submission
Saudi Arabia	Jeddah-2014	10,635	26/01/2014	KJ830751	AIH13925	Direct Submission

## Data Availability

All the data produced here is available and can produced when required.

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
