# Peer review of "Full-Length Genome and Partial Viral Genes Phylogenetic and Geographical Analysis of Dengue Serotype 3 Isolates"

_microorganisms, 2021, doi:10.3390/microorganisms9020323_

Round 1

Reviewer 1 Report

The authors have provided this revised manuscript and detailed their response to my initial concerns. However, I am not satisfied that they have done this adequately. Notably:

  • Could the authors provide some insight into why there might or might not be a preference for these genes to change?
    • The response was not sufficient nor included within the discussion. This needs to be amended
  • When considering the fidelity of the viral polymerase is this just evolution of the genome or variation due to other pressures on the virus?
    • The response was not sufficient nor included within the discussion. This needs to be amended
  • Is there a label missing on the far right brackets above the Oceania grouping and in Fig 2?
    • These may be cluster lines but it is unclear what they are actually showing or depicting. This needs to be further explained within the text and within the Figure Legends

Author Response

Subject: Submission of revised manuscript  microorganisms-1085288

Dear Reviewer,                   

It is stated that I want to submit the revised article entitled, “Full-length genome and partial viral genes phylogenetic and geographical analysis of Dengue serotype 3 isolates” for publication in Microorganisms Journal. We are highly thankful to referees whose comments helped in improving this manuscript. We have addressed most comments in uploaded version also. Below is response to referee comments:

Q: Could the authors provide some insight into why there might or might not be a preference for these genes to change?

Ans: The mutation rates not simply caused by polymerase errors, but virus corrects DNA mismatches by proofreading and/or post-replicative repair. Due to these qualms, viral mutation rates range between 10−8 and 10−4 (s/n/c), depending upon polymerase fidelity, imbalances in nt pools, template sequence/structure and post replicative repair etc.,

Q: When considering the fidelity of the viral polymerase is this just evolution of the genome or variation due to other pressures on the virus?

Ans.  The viruses has the ability to adapt as per need of hosts and environments. Its has the capacity to generate de novo diversity. Viral mutation rates is dependent upon polymerase fidelity, sequence/template structure, replication/post replication repair or host directed cytidine/adenine deaminases, in short it dependent upon number of virus and host dependent processes. Viruses presents high mutation rates in order to escape the immunity, e.g., cytotoxic T lymphocytes, antibody evasion and a number of point mutations are linked with vaccination failure and immune escape.

Q: Is there a label missing on the far-right brackets above the Oceania grouping and in Fig 2?

Ans. Labels are not missing, only cluster lines, to under the clusters or differentiate the two clusters. 

Q: I would still recommend them to do and present the statistical analysis of the phylogenetic tree to prove that the viral proteins they are investigating are indeed significant.

To understand the macroevolution needs temporal and topological methods. The temporal methods estimate diversification and topological methods are builds upon statistical measures of tree imbalance, which are two different approaches. For example, apTreeshape, for simulation and analysis of phylogenetic tree topologies using statistical imbalance. A separate manuscript may be required for analysis done (whole genome ( nt or AA) / gene(s)/protein(s)) in the context temporal macroevolution.

Thank you once again for your valuable comments. I am available if there are any further queries.

--

Best regards,

Prof. Dr. Marius Moga

Reviewer 2 Report

The authors have mostly addressed my comments. However, I would still recommend them to do and present the statistical analysis of the phylogenetic tree to prove that the viral proteins they are investigating are indeed significant.

Author Response

Dear Reviewer,                   

It is stated that I want to submit the revised article entitled, “Full-length genome and partial viral genes phylogenetic and geographical analysis of Dengue serotype 3 isolates” for publication in Microorganisms Journal. We are highly thankful to referees whose comments helped in improving this manuscript. We have addressed most comments in uploaded version also. Below is response to referee comments:

Q. Could the authors provide some insight into why there might or might not be a preference for these genes to change?

Ans: The mutation rates not simply caused by polymerase errors, but virus corrects DNA mismatches by proofreading and/or post-replicative repair. Due to these qualms, viral mutation rates range between 10−8 and 10−4 (s/n/c), depending upon polymerase fidelity, imbalances in nt pools, template sequence/structure and post replicative repair etc.,

Q. When considering the fidelity of the viral polymerase is this just evolution of the genome or variation due to other pressures on the virus?

Ans.  The viruses has the ability to adapt as per need of hosts and environments. Its has the capacity to generate de novo diversity. Viral mutation rates is dependent upon polymerase fidelity, sequence/template structure, replication/post replication repair or host directed cytidine/adenine deaminases, in short it dependent upon number of virus and host dependent processes. Viruses presents high mutation rates in order to escape the immunity, e.g., cytotoxic T lymphocytes, antibody evasion and a number of point mutations are linked with vaccination failure and immune escape.

Q. Is there a label missing on the far-right brackets above the Oceania grouping and in Fig 2?

Ans. Labels are not missing, only cluster lines, to under the clusters or differentiate the two clusters. 

Q. I would still recommend them to do and present the statistical analysis of the phylogenetic tree to prove that the viral proteins they are investigating are indeed significant.

To understand the macroevolution needs temporal and topological methods. The temporal methods estimate diversification and topological methods are builds upon statistical measures of tree imbalance, which are two different approaches. For example, apTreeshape, for simulation and analysis of phylogenetic tree topologies using statistical imbalance. A separate manuscript may be required for analysis done (whole genome ( nt or AA) / gene(s)/protein(s)) in the context temporal macroevolution.

Thank you once again for your valuable comments. I am available if there are any further queries.

--

Best regards,

Prof. Dr. Marius Moga

Reviewer 3 Report

The manuscript still needs extensive work of wording and presentation of data. The responses to the two reviewers are inadequate.

Author Response

Dear Reviewer,                   

It is stated that I want to submit the revised article entitled, “Full-length genome and partial viral genes phylogenetic and geographical analysis of Dengue serotype 3 isolates” for publication in Microorganisms Journal. We are highly thankful to referees whose comments helped in improving this manuscript. We have addressed most comments in uploaded version also. Below is response to referee comments:

Q. Could the authors provide some insight into why there might or might not be a preference for these genes to change?

Ans: The mutation rates not simply caused by polymerase errors, but virus corrects DNA mismatches by proofreading and/or post-replicative repair. Due to these qualms, viral mutation rates range between 10−8 and 10−4 (s/n/c), depending upon polymerase fidelity, imbalances in nt pools, template sequence/structure and post replicative repair etc.,

Q. When considering the fidelity of the viral polymerase is this just evolution of the genome or variation due to other pressures on the virus?

Ans.  The viruses has the ability to adapt as per need of hosts and environments. Its has the capacity to generate de novo diversity. Viral mutation rates is dependent upon polymerase fidelity, sequence/template structure, replication/post replication repair or host directed cytidine/adenine deaminases, in short it dependent upon number of virus and host dependent processes. Viruses presents high mutation rates in order to escape the immunity, e.g., cytotoxic T lymphocytes, antibody evasion and a number of point mutations are linked with vaccination failure and immune escape.

Q. Is there a label missing on the far-right brackets above the Oceania grouping and in Fig 2?

Ans. Labels are not missing, only cluster lines, to under the clusters or differentiate the two clusters. 

Q. I would still recommend them to do and present the statistical analysis of the phylogenetic tree to prove that the viral proteins they are investigating are indeed significant.

To understand the macroevolution needs temporal and topological methods. The temporal methods estimate diversification and topological methods are builds upon statistical measures of tree imbalance, which are two different approaches. For example, apTreeshape, for simulation and analysis of phylogenetic tree topologies using statistical imbalance. A separate manuscript may be required for analysis done (whole genome ( nt or AA) / gene(s)/protein(s)) in the context temporal macroevolution.

Thank you once again for your valuable comments. I am available if there are any further queries.

--

Best regards,

Prof. Dr. Marius Moga

This manuscript is a resubmission of an earlier submission. The following is a list of the peer review reports and author responses from that submission.

Round 1

Reviewer 1 Report

The authors attempt to compare whole genome of dengue virus serotype 3 across various genotypes and geographical regions to find out the similarities and differences. The main concern I have is the way they have presented the data, which show so many trees (Figure 4 and 5) that it makes it extremely difficult for the audience to appreciate the key findings of the paper. Also, the authors have not addressed precisely whether the phylogenetic differences observed have any effects on virus. Specifically, the main concerns are as follows:

  1. The authors used amino acid and nucleotide phylogeny on the same sample set, to show that the trend is the same (Figure 1 and 2). Isn't this expected since the nucleotide should translate to amino acid sequence, other than the 5' and 3' UTR regions?
  2. For the whole genome comparisons, especially the 5' UTR regions, are the sequences at the 5' end reliable? Many of the sequencing results rely on primers to sit at the ends of the DENV sequence so it is very likely that the 5' UTR first 13-15nt sequences are actually primer sequences.
  3. Figure 4-5 has many charts plotted but the authors have made no effort in digesting the information to put down the most critical trees that readers should be focusing on? Also, the comparisons are often very descriptive and there are no statistics to show whether particular DENV regions are significant or not. The authors should consider statistical tests such as the Shimodaira-Hasegawa and the KashinoHasegawa likelihood based tests (Gayathri Manokaran et al., Science, 2015) to quantify the significance. Without these statistical tests, the comparisons are purely descriptive.
  4. The authors will need to reference to existing literature to explain what the genome differences mean to viral fitness or epidemiological fitness, so as to inform readers what the genetic differences between genotypes mean.

Author Response

Dear Reviewer,

We thank you for your cooperation and we appreciate you taking the time to analyze our work. Following your comments and suggestions, we made some revisions to our paper. The comments regarding the changes we made can be found throughout the manuscript, and they are clearly highlighted.

Reviewer 2 Report

The authors conducted a phylogenetic and geographical study for Dengue virus. This could be an interesting study. However, the manuscript will strongly benefit from a major language editing. In the present form, the manuscript is not suitable for publication. 

Author Response

(The authors gave the same response as above.)

Reviewer 3 Report

The authors have analyzed the phylogenetic relationship between 33 GenBank accessed dengue 3 genomes. They have analyzed the relationship and sequence similarity and variability across a number of the dengue gene, particuarly those encoded non-structural proteins. The sequences themselves were submitted from various geographical locations. The authors utilized MEGA7 software and Bootstrap sampling as a basis for their conclusions. These being that there is a frequent divergence with the viral genes but some clear geographical clustering.

Some points to clarify:

  • In the analysis were there specific areas of the genome that were subject to greater variability than others? This was not clear in the report itself
  • Could the authors provide some insight into why there might or might not be a preference for these genes to change?
  • When considering the fidelity of the viral polymerase is this just evolution of the genome or variation due to other pressures on the virus?
  • Is there a label missing on the far right brackets above the Oceania grouping?
  • Similar question regarding the groupings indicated in Fig 2, are there meant to be labels on the brackets? What do these represent?

Author Response

(The authors gave the same response as above.)
